# The Fidelity of Rheumatoid Arthritis Multivariate Diagnostic Biomarkers Using Discriminant Analysis and Binary Logistic Regression

**DOI:** 10.3390/biom13091305

**Published:** 2023-08-25

**Authors:** Wail M. Hassan, Nashwa Othman, Maha Daghestani, Arjumand Warsy, Maha A. Omair, Eman Alqurtas, Shireen Amin, Abdulaziz Ismail, Afaf El-Ansary, Ramesa Shafi Bhat, Mohammed A. Omair

**Affiliations:** 1Department of Biomedical Sciences, University of Missouri-Kansas City School of Medicine, Kansas City, MO 64108, USA; hassanwm@umkc.edu; 2Central Research Laboratory, Center for Science and Medical Studies for Girls, King Saud University, Riyadh 11495, Saudi Arabia; nashwa_o@hotmail.com (N.O.); aswarsy@gmail.com (A.W.); 3Department of Zoology, College of Science, Center for Science and Medical Studies for Girls, King Saud University, Riyadh 11495, Saudi Arabia; mdaghestani@ksu.edu.sa; 4Department of Statistics and Operations Research, College of Sciences, King Saud University, Riyadh 11495, Saudi Arabia; maomair@ksu.edu.sa; 5Rheumatology Unit, Department of Medicine, College of Medicine, King Saud University, Riyadh 11495, Saudi Arabia; ealqurtas@ksu.edu.sa (E.A.); shireen.amin@hotmail.com (S.A.); momair@ksu.edu.sa (M.A.O.); 6College of Medicine, King Saud Bin Abdulaziz University for Health Sciences, Riyadh 11495, Saudi Arabia; ismail.abdulazizm@gmail.com; 7Biochemistry Department, College of Science, Center for Science and Medical Studies for Girls, King Saud University, Riyadh 11495, Saudi Arabia; rbhat@ksu.edu.sa

**Keywords:** autoimmune inflammatory, biomarkers, cytokines, chemokines, rheumatoid arthritis

## Abstract

Rheumatoid arthritis (RA) is an autoimmune inflammatory disease that causes multi-articular synovitis. The illness is characterized by worsening inflammatory synovitis, which causes joint swelling and pain. Synovitis erodes articular cartilage and marginal bone, resulting in joint deterioration. This bone injury is expected to be permanent. Cytokines play a prominent role in the etiology of RA and could be useful as early diagnostic biomarkers. This research was carried out at Riyadh’s King Khalid University Hospital (KKUH). Patients were enrolled from the Rheumatology unit. Seventy-eight RA patients were recruited (67 (85.9%) females and 11 (14.1%) males). Patients were selected for participation by convenience sampling. Demographic data were collected, and disease activity measurements at 28 joints were recorded using the disease activity score (DAS-28). Age- and sex-matched controls from the general population were included in the study. A panel of 27 cytokines, chemokines, and growth factors was determined in patient and control sera. Binary logistic regression (BLR) and discriminant analysis (DA) were used to analyze the data. We show that multiple cytokine biomarker profiles successfully distinguished RA patients from healthy controls. IL-17, IL-4, and RANTES were among the most predictive variables and were the only biomarkers incorporated into both BLR and DA predictive models for pooled participants (men and women). In the women-only models, the significant cytokines incorporated in the model were IL-4, IL-17, MIP-1b, and RANTES for the BLR model and IL-4, IL-1Ra, GM-CSF, IL-17, and eotaxin for the DA model. The BLR and DA men-only models contained one cytokine each, eotaxin for BLR and platelet-derived growth factor-bb (PDGF-BB) for DA. We show that BLR has a higher fidelity in identifying RA patients than DA. We also found that the use of gender-specific models marginally improves detection fidelity, indicating a possible benefit in clinical diagnosis. More research is needed to determine whether this conclusion will hold true in various and larger patient populations.

## 1. Introduction

Rheumatoid arthritis (RA) is a chronic autoimmune inflammatory disease characterized by multi-articular synovitis. The condition is distinguished by increasing inflammatory synovitis characterized by joint swelling and discomfort. Synovitis leads to joint degeneration by causing irreversible articular cartilage and marginal bone erosion [1].

Despite the plethora of therapeutic options and treatment algorithms currently available for RA, the accurate prediction of optimum treatment selection, course of disease, and mortality risk for individual patients remains elusive. Currently, RA clinical parameters and acute phase reactants (APRs) represent the core dataset that often guides therapeutic decisions for such a complex disorder as RA [2,3].

Circulating and/or synovial cytokines are regarded as excellent biomarkers for monitoring disease onset, development, and progression, either directly using cytokine multiplex immunoassays or indirectly by gene profiling approaches [4,5], because of the crucial role that cytokine networks play in maintaining inflammatory responses in the rheumatoid joint. However, only a small number of studies have employed circulating cytokine profiles to predict RA severity and outcome or to offer insights into its immune pathogenesis [4,5].

Interleukin-4 (IL-4) has recently been shown to play a function in the pathogenesis of rheumatism. IL-4 regulates the activation, differentiation, proliferation, and survival of several T-cell types. It was also shown to possess immunomodulatory properties in B cells, mast cells, macrophages, and a variety of other cell types [6]. IL-17A primarily acts on synoviocytes and osteoblasts to mediate cartilage and bone degradation in RA. In vitro experiments reveal that IL-17A causes synoviocyte activation (e.g., the production of IL-6 and IL-8), migration, and invasion, all of which increase cartilage degradation [7].

A negative correlation was suggested between genetically determined serum levels of macrophage inflammatory protein-1 (MIP-1b) and a lower risk of RA [8]. Interestingly, eotaxin 1 (CCL11) interacts with CCR3 to cause the migration of various leukocyte types. Fibroblast-like synoviocytes (FLSs), significant CCR3-expressing cells, are pathogenic effectors in RA. Wakabayashi et al. [9] reported that CCL11 levels in serum and synovial fluids (SFs) from RA patients were significantly higher than those in serum from healthy controls and SFs from osteoarthritis patients.

Wang et al. [10] emphasized the importance of platelet-derived growth factor (PDGF-BB) as a biomarker in RA. Fibroblast-like synoviocytes (FLSs) in the synovium of patients with RA can promote cartilage and bone destruction by producing matrix metalloproteinases and receptor activators of NF-κB ligands, thereby representing an important therapeutic target for RA. Furthermore, PDGF-BB plays an important role in RA-FLS proliferation and migration [10]. These results suggest that PDGF-BB might contribute to RA pathogenesis.

RA synovial fibroblasts (RASFs) release chemokines in response to interleukin-1 (IL-1) and tumor necrosis factor-alpha (TNF-α), which attract inflammatory cells to the site of inflammation [11]. Among these chemokines is the regulated upon activation, normal T expressed and presumably secreted (RANTES)/CC ligand 5 (CCL5), which is a potent CC chemokine demonstrated to be crucial in the pathophysiology of RA [12].

Granulocyte-macrophage colony-stimulating factor (GM-CSF) was first described as a growth factor that induced the differentiation and proliferation of myeloid progenitors in the bone marrow. It is interesting to highlight that mice with faulty GM-CSF are unable to develop arthritis in the collagen-induced arthritis (CIA) animal model, and antibodies against GM-CSF stop the disease from progressing and reduce proinflammatory cytokines in the joints [13].

This information sparked our interest in measuring a panel of cytokines in the plasma of RA patients and comparing them to those of healthy controls. Our goal is to help the early diagnosis of RA and identify therapeutic targets that are directly related to the severity of RA.

## 2. Materials and Methods

### 2.1. Patients and Controls

This study was carried out at King Khalid University Hospital (KKUH) in Riyadh. Ethical approval was obtained from the Ethical Committee (IRB No. E-14-1194). A convenient sampling criterion was developed to recruit consecutive patients from the Rheumatology unit. Seventy-eight RA patients were recruited (females: 67 (85.9%) and males: 11 (14.1%)). Demographics and disease activity measures using the disease activity score at 28 joints (DAS-28) were recorded. RA patients were grouped into high, moderate, and low based on pain, fatigue, inflammation, and remission groups. DAS-28 ≤ 2.6 corresponds to remission; >2.6 and ≤3.2 corresponds to low disease severity; DAS-28 ≥ 3.2 and ≤5.1 corresponds to moderate disease severity, while DAS ≥ 5.1 represents high disease severity [14]. Table 1 demonstrates the clinical and demographic data of RA patients and controls. Inclusion criteria were age ≥ 18 years and fulfilling the 2010 American College of Rheumatology (ACR) and the European League Against Rheumatism (EULAR) classification criteria [14] with disease duration ≥6 months. Exclusion criteria included systemic lupus erythematosus, systemic sclerosis, inflammatory myopathies, end-stage organ dysfunction, current or present diagnosis of malignancy, and pregnancy. Seventy-eight age- and sex-matched controls from the general population were included in the study. Written informed consent was obtained from each participant following the national protocol. Patients’ demographics, disease duration, seropositivity, and current therapies were extracted from medical records.

### 2.2. Sample Acquisition and Preparation

Blood was drawn by venipuncture, collected in EDTA tubes, and kept in ice boxes for transportation. The samples were immediately centrifuged, and plasma samples were separated according to instructions given by the Bio-Plex ProTM Human Cytokine Grp 1 Panel 27-Plex kit (BIO-RAD, Hercules, CA, USA). Aliquots of the plasma were stored at −80 °C until required for analysis.

The samples were thawed directly before the Bio-Plex assay using the Bio-Plex TM 200 system (BIO-RAD). Multiplex assays were chosen because they offer several advantages, including the ability to execute several reactions on a single sample and deliver extra information from the sample in a quick and efficient manner.

The Bio-Plex^®^ system is built around three core technologies. The first xMAP technology uses up to 100 unique fluorescently dyed beads, which permit the simultaneous detection of up to 100 different types of molecules in a single well of a 96-well microplate. The second is a flow cytometer with two lasers and associated optics to measure the different molecules bound to the surface of the beads. The third is a high-speed digital signal processor that efficiently manages fluorescent data.

The assay uses magnetic beads with anti-cytokine immunoglobulins to assess concentrations (pg/mL) of many cytokines simultaneously. The principle of the assay is similar to that of the sandwich ELISA, where the capture antibodies, directed to the target biomarker, are covalently coupled to the beads. The coupled beads react with the sample containing the biomarker of interest. After a series of washes to remove the unbound protein, a biotinylated detection antibody is added to create a sandwich complex. The final detection complex is formed with the addition of a streptavidin–phycoerythrin (SA-PE) conjugate. Phycoerythrin serves as a fluorescent indicator or reporter.

The determination of cytokines, chemokines, and growth factors was carried out in the patients and control samples in 4 runs. Each run consisted of a blank (sample diluent only), 8 standards (obtained from Bio-Rad), 20 RA patient samples, and 20 normal subjects (control) samples, all run in duplicates. The samples were processed following the manufacturer’s instructions (Bio-Plex Pro™ Human Cytokine Assays, Bio-Rad Laboratories), and the levels of the studied parameters were read using Bio-Plex Manager 5 Software.

### 2.3. Statistical Analysis

#### 2.3.1. Hierarchical Clustering

To perform hierarchical clustering, a similarity coefficient and a dendrogram (tree)-building algorithm are needed. The similarity coefficient is used to calculate all possible pairwise similarities in the dataset, while the algorithm constructs a tree with similar subjects clustered in common branches. Similarity in the current study was calculated using Canberra distances [15,16] (Equation (1)), and the Unweighted Pair Group Method with Arithmetic Mean algorithm [17] was used for building the trees. Hierarchical clustering was performed using Bionumerics version 6.6 (Applied Maths, Austin, TX, USA).
(1)$$D=\frac{1}{n}\sum_{i=1}^{n}\frac{\left|Xi-Yi\right|}{\left|Xi+Yi\right|}$$
where “*D*” is the Canberra distance metric, “*n*” is the number of variables, “*i*” is the *i*th variable, and “*X*” and “*Y*” are two participants.

#### 2.3.2. Binary Logistic Regression

Binary logistic regression (BLR) was used to predict RA among participants. BLR uses a linear combination of multiple independent variables to predict a dichotomous categorical variable [18]. The BLR model is trained using the natural log of odds, and the odds can be calculated from the known classification of participants in the dataset using Equations (2) and (3). The regression process is based on the relationship between the natural log of odds and the predictor variables. The significance of the model is tested using a chi-square test with *p*-values < 0.05 considered significant. The model is further evaluated using the Hosmer–Lemeshow test and Nagelkerke’s pseudo-*R^2^*. The Hosmer–Lemeshow test predicts classification accuracy, which is assumed by retaining the hypothesis of perfect prediction with *p*-values > 0.05 [19]. Nagelkerke’s pseudo-*R^2^* can take values between 0 and 1 and indicates the proportion of explained variance in the dependent variable [20]. The importance of each independent variable to the model was evaluated in multiple ways. First, the Wald chi-square test was used to test the significance of the variable’s contribution to the model, with *p*-values < 0.05 indicating significance. Second, the amount of contribution of each variable was evaluated using the variable’s coefficient and odds ratio. The coefficient is the weight assigned to each independent variable in the logistic regression equation, with more important variables having greater coefficients. The odds ratio is the predicted variation in the dependent variable caused by an independent variable’s unit change with all other independent variables remaining unchanged.

Equation (2) is the probability of falling in a target group (*P_t_*) (e.g., rheumatoid arthritis):(2)$$Pt=\frac{\mathrm{Number}\;\mathrm{of}\;\mathrm{participants}\;\mathrm{in}\;\mathrm{the}\;\mathrm{target}\;\mathrm{group}}{\mathrm{Total}\;\mathrm{number}\;\mathrm{of}\;\mathrm{participants}}$$

Equation (3) is the relationship between the odds of falling in a target group, the probability of falling in that group (*P_t_*), and the probability of falling in the alternate group (*P_a_*):(3)$$\mathrm{Odds}=\frac{Pt}{Pa}=\frac{\frac{\mathrm{Number}\;\mathrm{of}\;\mathrm{participants}\;\mathrm{in}\;\mathrm{the}\;\mathrm{target}\;\mathrm{group}}{\mathrm{Total}\;\mathrm{number}\;\mathrm{of}\;\mathrm{participants}}}{\frac{\mathrm{Number}\;\mathrm{of}\;\mathrm{participants}\;\mathrm{in}\;\mathrm{the}\;\mathrm{alternate}\;\mathrm{group}}{\mathrm{Total}\;\mathrm{number}\;\mathrm{of}\;\mathrm{participants}}}=\frac{\mathrm{Number}\;\mathrm{of}\;\mathrm{participants}\;\mathrm{in}\;\mathrm{the}\;\mathrm{target}\;\mathrm{group}}{\mathrm{Number}\;\mathrm{of}\;\mathrm{participants}\;\mathrm{in}\;\mathrm{the}\;\mathrm{alternate}\;\mathrm{group}}$$

Finally, a model’s performance was empirically tested by determining its success rate in correctly classifying participants into either the RA group or the control group. To avoid potentially overinflating the rate of correct classification due to attempting to classify the same subjects used to train the model, the model was recalculated before attempting to classify each of the participants with the participant being classified excluded. BLR was performed using IBM SPSS version 26 (IBM Corporation, Armonk, NY, USA).

#### 2.3.3. Discriminant Analysis

Discriminant analysis (DA) combines correlated variables into new vectors known as canonical discriminant functions. The discriminant function is a selection process that aims to maximize group separation [18]. The discriminant models were evaluated by the Wilks’ λ statistic, which represents the proportion of unexplained variance and ranges from 0 to 1. Therefore, better models have lower Wilks’ λ values [18]. The significance of the model was evaluated with a chi-square test with *p*-values lower than 0.05 indicating significance [18]. DA results were plotted using GraphPad Prism version 6 for Windows (GraphPad Software, San Diego, CA, USA, www.graphpad.com). Since we had two groups, there was only one discriminant function (number of discriminant functions = number of groups − 1). To plot the data with the sole canonical discriminant function represented on the x axis, we used arbitrary units on the y axis to clearly illustrate group segregation. Discriminant functions were evaluated according to the corresponding eigenvalue, which reflects the amount of variance explained [21], and canonical correlation, which reflects the discriminant function’s correlation with the groups [22]. The significance of independent variables’ contributions was evaluated using one-way ANOVA with a significance threshold of *p* < 0.05. Individual variables were also evaluated using the Wilks’ λ statistic, where a value of 0 implied that all variance in the independent variable could be explained by differences in group membership and a value of 1 implied that none of the variance could be explained by group membership [18]. The direct contribution of the variable to the discriminant model was reflected in the standardized canonical discriminant function coefficient [23], which was the main criterion in evaluating the importance of individual independent variables to the discriminant models. The rate of correct classification was tested as described for BLR. DA was performed using IBM SPSS version 26 (IBM Corporation, Armonk, NY, USA).

#### 2.3.4. Receiver Operating Characteristic Curve (ROC) and Area under the Curve (AUC)

The ROC curve is generated by plotting the rate of true positives (sensitivity) against the rate of false positives (1 − specificity) for all possible threshold values. For a test with 100% sensitivity and 100% specificity, the AUC is equal to 1, while an absolutely useless test has an AUC of 0.5. The asymptotic significance of the AUC is evaluated by testing the null hypothesis stating that the test has an AUC of 0.5. The null hypothesis is rejected when the adjusted *p*-value is <0.05. We also evaluated the models and individual biomarkers in terms of sensitivity and specificity at thresholds that correspond to approximately 80–90% sensitivity. ROC/AUC analyses were performed using IBM SPSS version 26 (IBM Corporation, Armonk, NY, USA).

#### 2.3.5. Other Statistical Testing

Correlations between RA probabilities, which were calculated by either BLR or DA, and clinical findings such as ESR, DAS28ESR, and anti-cyclic citrullinated peptide (anti-CCP) were estimated using Spearman correlations. Spearman correlations were performed using GraphPad Prism version 6 for Windows (GraphPad Software, San Diego, CA, USA, www.graphpad.com).

## 3. Results

Hierarchical clustering was employed to investigate unsupervised group partitioning among participants (Figure 1). A clearly discernable partitioning between RA participants and controls was observed whether the analysis was applied to all participants in one group or separately applied to men and women. We then proceeded to construct models designed to optimize the separation between patients and controls, for which we used BLR and DA.

### 3.1. Constructing Models to Distinguish RA Patients from Normal Controls Using BLR and DA

BLR and DA models were constructed from the data of all participants, only men, or only women. All six models were statistically significant, as indicated by *p*-values < 0.05 (Table 2 and Table 3). BLR models explained 53.5%, 63.2%, and 60.1% of data variance (Nagelkerke pseudo-R^2^) for the all-participants, men, and women models, respectively (Table 2). The Hosmer–Lemeshow *p*-value was significant (0.019) for the all-participants model, indicating that the model’s fit to the data is suboptimal. Conversely, gender-specific models were both well calibrated, as indicated by Hosmer–Lemeshow *p*-values > 0.05 (Table 2). As for the discriminant models, the women’s model was superior to the other two in multiple ways. First, the women’s model was the most correlated with group membership, as indicated by its canonical correlation (Table 3). Canonical correlation is the best measure of model fitness given the binary nature of the grouping variable (i.e., RA versus control). In addition, the eigenvalue and proportion of variance that can be explained by group membership (Wilks’ lambda − 1) were both highest in the women’s model (Table 3).

Since we had two groups, the RA and control groups, only one discriminant function was extracted. Plotting participants from both groups on the discriminant function of the all-participants model showed partial separation between groups (Figure 2). Participants were also plotted according to the probability of falling in the RA group calculated from the BLR models, which showed better separation compared to DA, especially among female participants (Figure 2).

The results of both BLR and DA pointed to IL-4 as the most important variable for the all-participants and women’s models. Specifically, in the BLR models of all participants and women, IL-4 had the greatest regression weights (1.387 and 1.548, respectively), and in the discriminant models, it had the greatest standardized canonical discriminant function coefficient (SCDFC) values (1.290 and 1.252) and the highest proportion of variance explained by group membership (i.e., the lowest Wilks’ lambda values: 0.743 and 0.710, respectively). Furthermore, the BLR models for all participants and women indicated that a unit increase in IL-4 results in a 4-fold or more increase in the odds of having RA (odds ratios of 4 and 4.7 for the all-participants and women models, respectively) (Table 4). 

It is noteworthy that, regardless of whether BLR or DA was used, the variables in the model and their order of importance were identical in the all-participants models. In addition to IL-4, IL-17, and RANTES appeared to significantly improve the models, which is likely to augment the predictive power of the models. The women’s models also contained more variables in addition to IL-4. The BLR model had IL-17, MIP-1b, and RANTES in descending order of importance, while the discriminant model had IL-1Ra, GM-CSF, IL-17, eotaxin, and RANTES (Table 4). PDGF-BB was the only variable in the discriminant model for men and was the second most important factor in the BLR model after eotaxin (Table 4). It is noteworthy that both of the men’s models (BLR and DA) were weak. For example, the BLR model had variable weights close to 0 and odds ratios approaching 1 (Table 4), which meant that variable contributions were minimal and changes in the predictors did not result in sizable changes in the odds of having rheumatoid arthritis, respectively.

### 3.2. Testing the Predictive Power of the Models

To directly test the predictive power of each model, we compared the rates of correct classification of the six models. We show that BLR models outperformed their corresponding DA models, slightly for the all-participants and female models and dramatically for the male models (Figure 3).

We then wanted to compare the predictive power of each model to the predictive power of each of its constituent variables, which was accomplished using ROC curves and AUCs. Using the AUCs, we were also able to evaluate the models and compare them to each other using a technique that is computationally independent of BLR and DA. As expected from the quality of the models described above, the all-participants and women models were superior to the men’s models. Although the AUCs obtained by the men’s models, BLR and DA, were comparable to those obtained by the other models, the specificity was generally inferior to other models at thresholds corresponding to 82–91% sensitivity (Figure 4 and Table 5). At 90.9% sensitivity, the specificity of the men’s BLR and DA models were both equal to 45.5%. At 81.8% sensitivity, these two models had 100% and 63.5% specificity, respectively. Despite the perfect specificity obtained using the men’s BLR model at the lower sensitivity of 81.8%, which is the highest among all other models, such results should be interpreted with caution given the model’s quality and very small sample size. That is also in addition to the low specificity obtained with men’s models, including the BLR model at a sensitivity of 90.9%. For example, the BLR models of all participants and women showed specificities of 82.1% (sensitivity 89.6%) and 83.3% (sensitivity 85.9%), respectively (Table 5). A similar trend was also observed for the DA models (Table 5). The obtained ROC-AUCs, specificity, and sensitivity of the measured panel of cytokines are comparable to those recorded in other studies, which confirms the possibility of using our models as predictive biomarkers for the early diagnosis of RA (Table 5).

### 3.3. Estimated Probability of RA Correlates with Clinical Findings

The probability of RA can be calculated from both BLR and DA models. To test the clinical applicability of our models, we asked whether these probabilities correlated with common clinical findings in RA patients, namely ESR, DAS28ESR, and anti-CCP. Since we had clinical data for 57 RA patients, 55 women and 2 men, we performed the analysis on the women group and the entire group of 57 patients. With only two men, we could not perform a correlation analysis on the men group. We show that the probabilities of RA performed on the same population using BLR or DA were highly correlated (r = 0.987 and 0.824 for all patients and female patients, respectively) (Figure 5 and Appendix A). The RA probabilities among the entire group correlated the most with ESR (r = 0.350 and 0.339 using DA and BLR, respectively), although the correlation was weak. Probabilities calculated using DA also significantly, but weakly, correlated with DAS28ESR (r = 0.262) (Figure 5). The probability of RA among female patients correlated significantly, but also weakly, with anti-CCP (r = 0.355 and 0.328 using DA and BLR, respectively); when calculated using DA, it also correlated with ESR (r = 0.324) (Appendix A).

## 4. Discussion

This study demonstrated the significance of using a panel of cytokines as a cost-effective multivariate biomarker for the early detection of RA. Early diagnosis and intervention are needed to help avoid the serious complications of RA and improve patients’ quality of life [14]. Although further research is needed to confirm the utility of our methodology for clinical practice, our data suggest that our approach is promising enough to warrant further investigations. We show that the data have enough variance contributing to group separation to noticeably segregate patients from controls using hierarchical clustering—an unsupervised partitioning method—as well as using methods that maximize group separation such as BLR and DA. We have also shown clinically acceptable sensitivity and specificity, as well as rates of correct classification, with the all-participants and women models. We have also shown that the use of multivariate biomarker profiles improves sensitivity and specificity over the use of individual biomarkers; the latter is typically insufficient to predict the prognosis of RA [24].

The role of cytokines, including the ones making up our BLR and DA models, in RA pathogenesis has been well documented in the literature [25]. Cytokines are particularly important for the emergence of both arthritic and systemic RA symptoms, as well as for the conversion of acute inflammation into a chronic process [26,27]. Bone homeostasis is maintained by bone resorption and formation by osteoclasts and osteoblasts, respectively [28]. The osteoclast, which plays a critical role in the initiation of joint and bone erosion in RA, is a known target of numerous cytokines, including ones that are osteoclastogenic (e.g., IL-1 and IL-17) and others that are anti-osteoclastogenic (e.g., IL-4) [28]. Therefore, the use of cytokine biomarker profiles to enable the accurate and early diagnosis of RA is rational. 

Our results underscore the importance of a few cytokines in separating RA from healthy controls. IL-4 was the most important among these cytokines, which is consistent with multiple previous reports. RA patients were found to have elevated IL-4 mRNA levels in whole blood and elevated IL-4 protein in mononuclear and whole blood cells [29,30]. RA patients were also shown to have elevated IL-4 levels in synovial fluid and plasma [31,32]. Additionally, Kokkonen et al. [33] demonstrated elevated IL-4 levels in RA patients prior to the onset of the disease. We have also shown that IL-17 was one of the most influential biomarkers in separating RA patients and controls. The role of IL-17 in RA pathogenesis has been supported by multiple previous reports. IL-17 was found in the synovial fluids and membranes of RA patients, implicating Th17 cells in RA pathogenesis [34]. IL-17 is a proinflammatory cytokine that promotes bone resorption, which further implicates it in RA pathogenesis [34]. Since IL-17 promotes inflammation, blood coagulation, and thrombosis in arteries and cardiomyocytes [35], inhibiting IL-17 expression or reducing its interaction with its cognate receptor could become beneficial in treating cardiovascular disease in RA patients [36]. In fact, inhibiting IL-17 reduced inflammation and joint erosion in animal models [37]. Furthermore, Jain et al. [38] and Magda et al. [39] found a substantial association between IL-17 and DAS-28, which is consistent with IL-17 being a valuable biomarker of disease activity in RA. RANTES has also been linked to RA. Yao et al. [40] and Yang et al. [41] found that plasma RANTES levels in RA were more strongly correlated with disease activity than CRP or ESR. They also showed that low RANTES plasma levels were conversely correlated with the duration of clinical remission in RA [40,41], indicating that plasma RANTES can be utilized as a predictor of RA activity.

The involvement of PDGF-BB is supported by numerous studies that showed elevated PDGF signaling, proliferation, and accumulation of fibroblasts that express the PDGF receptor (PDGFR) in the synovial fluid of RA patients. An active form of PDGFR known as phosphorylated PDGFR (pPDGFR) has been found in abnormally high concentrations in the synovium of RA patients [42,43,44,45]. The DA model of males contained PDGF-BB as a sole biomarker, indicating that there was no other biomarker that could have significantly improved the model’s predictive power. Since the DA model contains PDGF-BB as a sole biomarker, the AUC of the model (0.818) was equal to that of PDGF-BB alone. However, the men’s models were weak, as discussed in the results section, a fact that could have resulted from the small sample size of the men’s group. Another possible explanation is the difference in disease severity between men and women. Women with RA typically have a more aggressive disease, higher levels of disease activity, and higher incidences of disability along with less improvement in disability and the highest radiographic progression using the Sharp van der Heijde Score (SHS), primarily because of increased joint space narrowing [46,47]. This suggests a potential benefit in using gender-specific models to improve detection reliability, as recently recommended by Pertsinidou et al. [48] Considering the high AUCs of DA and PDGF-BB, however, it is tempting to speculate that the low qualities of the men’s models were more likely caused by the small sample size than by an intrinsic difficulty in the distinction between male RA patients and controls due the former’s relatively mild symptoms.

To further support our approach, we show correlations between the probabilities of having RA, as calculated by BLR and DA, and some of the most commonly used laboratory markers of RA, namely ESR, DAS28ESR, and anti-CCP. As shown in the results section, these were weak correlations that were inconsistent across models. Two of the most influential cytokines in the best two models we had, the all-participants model and the women’s models, were IL-4 and IL-17. In our study, IL-4 weakly correlated with ESR and anti-CCP, but not DAS28ESR, while IL-17 did not significantly correlate with any of the three laboratory tests. Both agreeing and conflicting results can be found in the literature. For example, one study reported that IL-17, but not IL-4, was moderately correlated with DAS28ESR and ESR [49], while another reported that IL-17 did not correlate with either DAS28 [50] or ESR [50,51]. These discrepancies may have resulted from differences between study populations (e.g., regarding age, disease severity, disease stage, proportion of males and females, genetic background, comorbidities, or medications), technical differences (e.g., regarding medical practices or data collection and analysis), or a combination of both. Furthermore, as we attempt to reconcile existing data and conceptualize future experiments aiming to decisively resolve such discrepancies, it is prudent to comprehend the highly complex nature of the mechanisms that govern such relationships. For instance, ESR increases during the acute phase response chiefly due to enhanced rouleau formation, which is dependent on the concentration of acute phase proteins and immunoglobulins in circulation, as well as the shape and density of erythrocytes in the bloodstream [52]. Fibrinogen is one of the acute phase proteins [52], meaning elevated fibrinogen levels in plasma should enhance rouleau formation and increase ESR. However, fibrinogen and ESR are elevated in RA [53], but in vitro experiments suggest that IL-4, which is also elevated in RA, inhibits the production of fibrinogen [54]. Taken together, although we could not fully explain the inconsistent, weak correlations between the probability of RA prediction based on our algorithms and conventional laboratory markers of RA, such findings do not surprise us given the complexity of the underlying mechanisms. Furthermore, the use of multivariate biomarker profiles may overcome the variability of individual biomarkers.

## 5. Conclusions

We demonstrated in this study that a cytokine-based, multiparametric biomarker profile successfully distinguished RA patients from healthy controls in a pool of 78 participants. This conclusion shows that the panel could be used as a diagnostic tool. We demonstrated that using BLR led to a higher fidelity of RA patients’ identification than using DA. Using gender-specific models slightly improved detection fidelity, indicating a potential benefit in clinical diagnostics. Further research is needed to show whether these findings will remain applicable using different and larger patient populations.

## Figures and Tables

**Figure 1 biomolecules-13-01305-f001:**
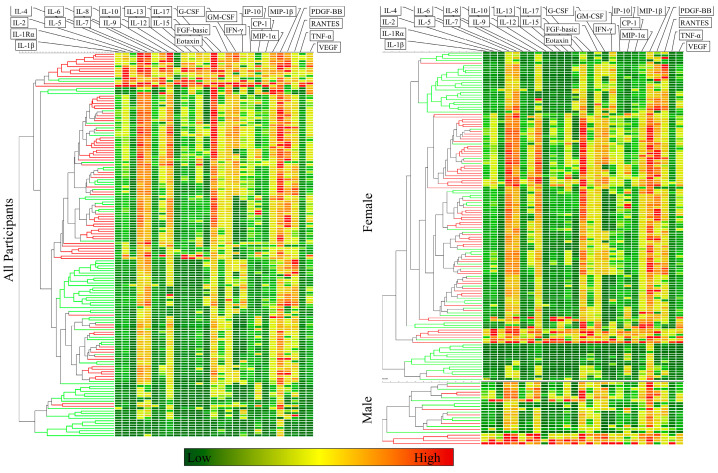
Unsupervised group partitioning using hierarchical clustering. Tree branches are colored according to participant status with RA in red and controls in green. The heat map shows the relative plasma concentration of analytes with elevated levels in red and low levels in green.

**Figure 2 biomolecules-13-01305-f002:**
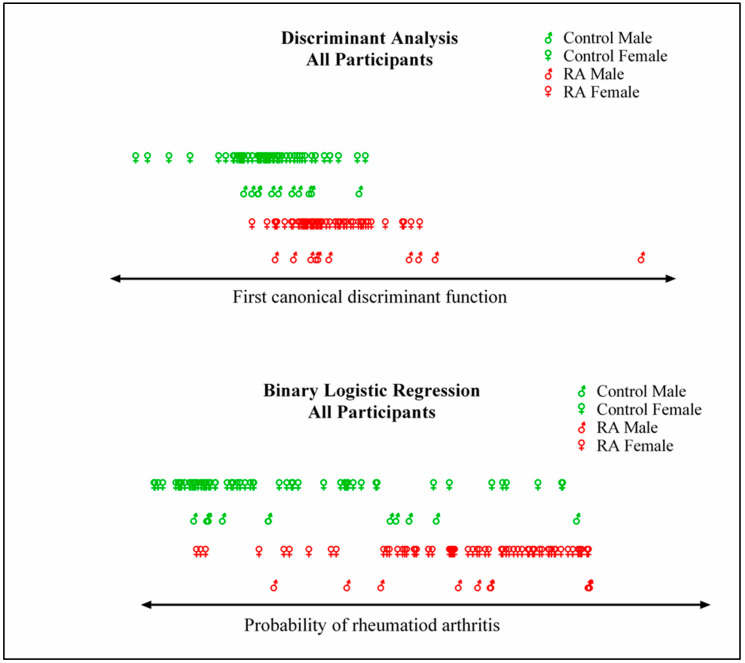
Separation of RA and controls. Participants were separated on one axis (the x axis) using either discriminant analysis or binary logistic regression. The variables used in this analysis included age, gender, and 27 plasma cytokines and growth factors. In the discriminant analysis plot (top), the x axis represents the first canonical discriminant function, and participants are graphed using their discriminant scores. In the binary logistic regression plot (bottom), the x axis represents the probability of having rheumatoid arthritis. Sample size: n = 78 each, control and RA; n = 22 (50% RA) males and 134 (50% RA) female participants.

**Figure 3 biomolecules-13-01305-f003:**
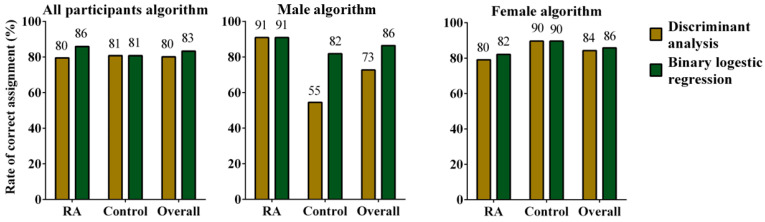
Rate of correct assignment (RCA). RCAs were determined using algorithms driven from the data collected from all participants (n = 78 each, control and rheumatoid arthritis), male participants (n = 11 each, control and rheumatoid arthritis), or female participants (n = 67 each, control and rheumatoid arthritis). Algorithm derivation was accomplished using either discriminant analysis or binary logistic regression. Algorithms were used to assign participants to either the RA group or control group within the corresponding participant population from which they were driven.

**Figure 4 biomolecules-13-01305-f004:**
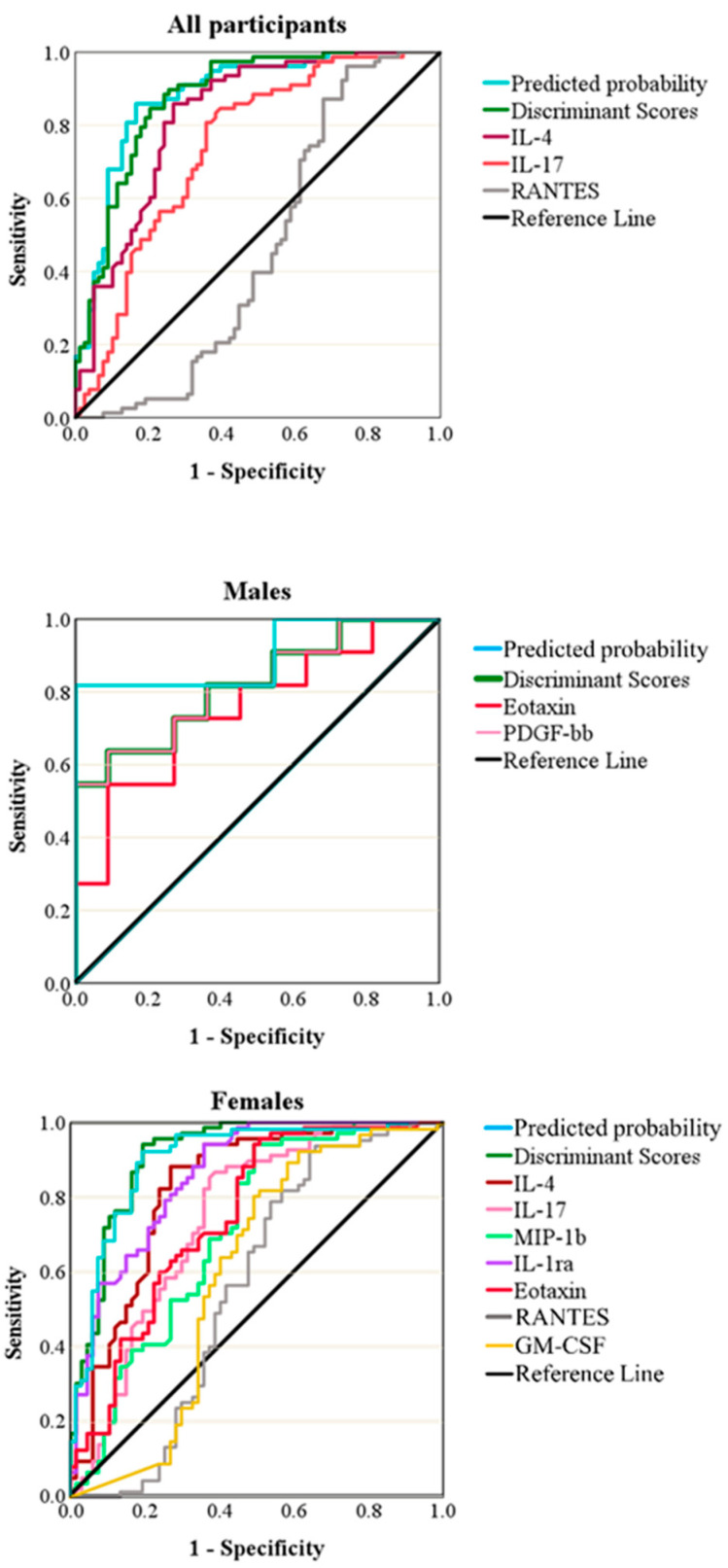
Receiver operating characteristic (ROC) curves for all participants, males, and females.

**Figure 5 biomolecules-13-01305-f005:**
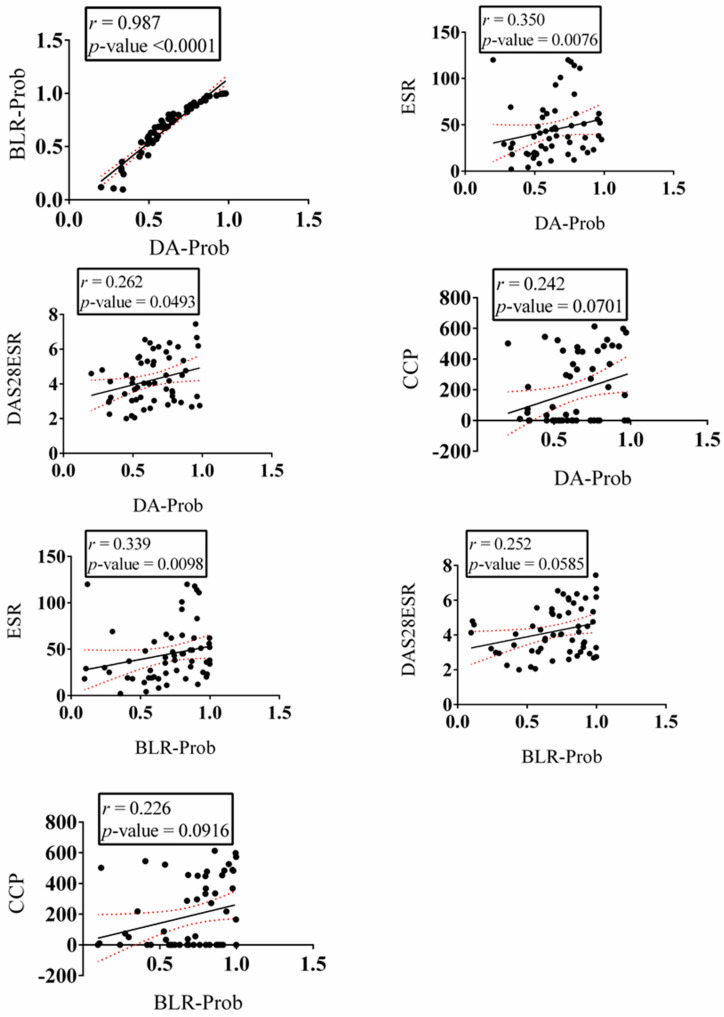
Estimated probability of RA diagnosis correlates with erythrocyte sedimentation rate (ESR), disease activity score-28 with erythrocyte sedimentation rate (DAS28ESR), and cyclic citrullinated peptide antibody (Anti-CCP). Analysis performed on male and female rheumatoid arthritis patients with available clinical data (n = 57). DA-Prob: probability estimated using discriminant analysis, BLR-Prob: probability estimated using binary logistic regression.

**Table 1 biomolecules-13-01305-t001:** Clinical and demographic data of patients and controls.

Variable	Number
Mean Age (years)	49.5 + 13.8
Female gender (%)	58 (87.8)
Duration of disease	
Early (≤2 years) Established (>2 years)	6 51
Seropositivity (%) (0 seronegative, 1 positive to RF or anti-CCP, 2 double positive) For RF and anti-CCP the cutoff for positive was 15 U/mL and 5 IU/mL, respectively.	
Rheumatoid factor (positive)	36 (63.15)
Anti-cyclic citrullinated peptide	49 (85.96)
0	15 (26.31)
1	16 (28.07)
2	35 (61.4)
Medications	
Drug-free remission	7 (12.28)
MTX monotherapy	11 (19.29)
MTX combination base	2 (3.5)
Other csDMARDs	5 (8.77)
bDMARDs/tsDMARDs	41 (71.92)
Controls	
Mean age (years)	47 + 14.4
Female gender (%) Female	68 (87.2)

N.B. Clinical data were available for 57 out of 78 RA patients.

**Table 2 biomolecules-13-01305-t002:** Fit of the binary regression models.

Measure of Model Fit or Significance	All Participants	Male Participants	Female Participants
Chi-square *p*-value	2.98 × 10^−17^	0.00086	1.55 × 10^−16^
Nagelkerke pseudo-R^2^	0.535	0.632	0.601
Hosmer–Lemeshow *p*-value	0.019	0.609	0.258

**Table 3 biomolecules-13-01305-t003:** Fit of the discriminant models.

Measure of Model Fit or Significance	All Participants	Male Participants	Female Participants
Canonical correlation	0.564	0.524	0.688
Eigenvalue	0.465	0.378	0.899
Wilks’ lambda	0.682	0.726	0.527
*p*-value	1.37 × 10^−12^	0.012	9.80 × 10^−16^

**Table 4 biomolecules-13-01305-t004:** Relative importance of individual biomarkers (variables) to each of the multivariate biomarker profiles (models). Individual biomarkers are ranked in descending order of their relative importance. In DA, the standardized canonical discriminant function coefficient (SCDFC) is the best indicator of relative importance, which is the basis of ranking on the left side of this table. Another measure of variable contribution to the discriminant model is Wilks’ lambda (λ), which indicates the amount of variance not explained by group membership. Smaller values of λ indicate better ability to discriminate between groups. In BLR models, the raw score regression coefficients (B) and odds ratios (Exp(B)) are used to rank the relative importance of individual biomarkers in a multivariate model. The *p*-values are obtained using one-way ANOVA to evaluate the equality of group means with values less than 0.05 considered significant.

Rank	Binary Logistic Regression Models			Discriminant Analysis Models		
	All Participants	Male	Female	All Participants	Male	Female
1	IL-4 B = 1.387 Exp(B) = 4.001 *p* = 3.97 × 10^−8^	Eotaxin B = 0.037 Exp(B) = 1.037 *p* = 0.288	IL-4 B = 1.548 Exp(B) = 4.703 *p* = 6.79 × 10^−8^	IL-4 SCDFC = 1.290 λ = 0.743 *p* = 1.46 × 10^−11^	PDGF-BB SCDFC = 1.000 λ = 0.726 *p* = 0.012	IL-4 SCDFC = 1.252 λ = 0.710 *p* = 1.87 × 10^−11^
2	IL-17 B = −0.027 Exp(B) = 0.974 *p* = 8.09 × 10^−4^	PDGF-BB B = 0.007 Exp(B) = 1.007 *p* = 0.070	IL-17 B = −0.038 Exp(B) = 0.963 *p* = 1.23 × 10^−4^	IL-17 SCDFC = −0.479 λ = 0.902 *p* = 7.09 × 10^−5^	n/a	IL-1Ra SCDFC = 0.783 λ = 0.900 *p* = 1.91 × 10^−4^
3	RANTES B = 0.000 Exp(B) = 1.000 *p* = 5.85 × 10^−4^	n/a	MIP-1b B = 0.054 Exp(B) = 1.055 *p* = 0.017	RANTES SCDFC = −0.439 λ = 0.967 *p* = 0.024	n/a	GM-CSF SCDFC = −0.598 λ = 0.943 *p* = 0.005
4	n/a	n/a	RANTES B = 0.000 Exp(B) = 1.000 *p* = 1.25 × 10^−4^	n/a	n/a	IL-17 SCDFC = −0.535 λ = 0.909 *p* = 3.85 × 10^−4^
5	n/a	n/a	n/a	n/a	n/a	Eotaxin SCDFC = −0.533 λ = 0.915 *p* = 6.20 × 10^−4^
6	n/a	n/a	n/a	n/a	n/a	RANTES SCDFC = −0.334 λ = 0.958 *p* = 0.017

**Table 5 biomolecules-13-01305-t005:** Predictive power evaluation using the area under a receiver operating characteristic curve (AUC) method.

Patient Population	Model/Variable	AUC	*p*-Value	Sensitivity	Specificity
All	BLR	0.882	1.76 × 10^−16^	85.9%	83.3%
	DA	0.877	4.14 × 10^−16^	85.9%	75.6%
	IL-4	0.825	2.37 × 10^−12^	85.9%	73.1%
	IL-17	0.742	1.85 × 10^−7^	85.9%	56.4%
	RANTES	0.463	0.421	85.9%	32.1%
Males	BLR	0.901	0.001	90.9% 81.1%	45.5% 100%
	DA	0.818	0.011	90.9% 81.1%	45.5% 63.5%
	PDGF-BB	0.818	0.091	90.9% 81.1%	45.5% 63.5%
	Eotaxin	0.752	0.045	90.9% 81.1%	36.4% 54.5%
Females	BLR	0.903	7.85 × 10^−16^	89.6%	82.1%
	DA	0.913	1.61 × 10^−16^	89.6%	80.6%
	IL-4	0.825	8.83 × 10^−11^	88.1%	73.1%
	IL-17	0.743	1.00 × 10^−6^	89.6%	52.2%
	MIP-1b	0.709	3.10 × 10^−5^	89.6%	50.7%
	IL-1Ra	0.860	6.33 × 10^−13^	89.6%	64.2%
	Eotaxin	0.747	7.66 × 10^−7^	89.6%	50.7%
	RANTES	0.562	0.219	89.6%	35.8%
	GM-CSF	0.597	0.052	89.6%	41.8%

## Data Availability

Presented in the manuscript.

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
