# Peer review of "The Fidelity of Rheumatoid Arthritis Multivariate Diagnostic Biomarkers Using Discriminant Analysis and Binary Logistic Regression"

_biomolecules, 2023, doi:10.3390/biom13091305_

Round 1
Reviewer 1 Report
The manuscript titled: “Multivariate diagnostic biomarkers of rheumatoid arthritis.” submitted to the journal Biomolecules with manuscript ID: biomolecules-2535160 by Wail M. Hassan et al. analyzed 27 serological factors by multiplex assay - Bio-Plex assay in a group of 78 cases with rheumatoid arthritis (RA). The authors analyzed data by different statistical tests and demonstrated that using binary logistic regression led to higher fidelity of rheumatoid arthritis patients’ identification than using discriminant analysis.
Although the general aim and scope of the journal are focused on structures and functions of bioactive and biogenic substances, molecular mechanisms with biological and medical implications as well as biomaterials and their applications, with the respect of the Bioinformatics and Systems Biology Section of the journal I accept that the manuscript could be of interest to the journal’s readers.
However, the manuscript has several weaknesses that should be taken into consideration.
Major issues:
Title:
1. The title “Multivariate diagnostic biomarkers of rheumatoid arthritis” should be revised.
In my opinion, the main topic of the manuscript is the research for the most trustful and powerful analysis able to discriminate individuals with RA from controls.
Abstract:
2. Page 1, lines 31-33: Please revise the sentence: “The obtained data showed that a 27-cytokine, multiparametric biomarker profile can successfully distinguish rheumatoid arthritis patients from healthy controls.”
3. Out of 27 tested serological biomarkers, just 4 or 5 of them (IL-4; IL-17; MIP-1b; RANTES; PDGF-BB) showed some significance. This should be clearly stated.
Materials and Methods:
2.1. Patients and controls
4. The small sample size. The study included 78 RA cases, 57 of which with available clinical data.
5. A detailed table with clinical and demographic data of cases and controls should be added. The current therapies also should be taken into account.
6. It is not clear how many controls were included. What inclusion and exclusion criteria were used for controls?
7. It is not clear for which analyses the RA patients were grouped into high, moderate, and low, and remission groups. The subdivision into 4 subgroups of RA patients probably will influence the power of the analysis.
8. A new subheading “Bio-Plex assay” or similar should be added. The procedure does not fit well with 2.2. “Sample acquisition and preparation”. The technical characteristics such as the sensitivity, precision, accuracy, detection range etc. of the assay should be added.
9. I may suggest validation of the obtained results, at least for the most significant cytokines, by the singleplex assays, as conventional ELISA tests that may strengthen your results. Multiplex approaches usually are accepted as screening tests, for the future focusing on the most significant parameters.
2.3. Statistical analysis
10. This part should be shortened and revised. The used reference in that part did not correspond to the reference list. Please, check all of them carefully. Also, correct the format of references according to the journal instruction.
Results and discussion:
11. The section “Results and discussion” should be separated into two “Results” and “Discussion”
12. The observed levels in pg/ml (row data, mean, or median) of studied cytokines, chemokines, and growth factors should be added.
13. Figure 2, should be revised. It is not entirely informative. What is presented on the x-axis? Also, please check the legend – the last sentence should be deleted.
14. Table 4 and Figure 4 could be combined into one.
15. The observed correlations with clinical parameters even significant are weak and this should be clearly stated.
16. Please check carefully the data presented in Figure 5 and Figure S1. For me, it is hard to believe that the differences between them are just 2 men, RA patients.
17. Discussion should be a separate part of the paper and should be focused on the obtained results.
18. How statements as on page 7, lines 310-313 “In blood arteries and cardiomyocytes, IL-17 promotes inflammation, blood coagulation, and thrombosis [25]. It is interesting to mention that Inhibiting IL-17 expression or reducing its interaction with the receptor may thus be beneficial in treating cardiovascular disease in RA patients [26].” is connected with the presented data. If cases with cardiovascular disease are included in the study, please clarify this.
19. Please, clarify the statement “…..ESR is a reflection of fibrinogen level in the blood………” (page 10, line 410)
20. For me it is not clear, how the associative study by Pawlik et al [40] concerning promoter polymorphism in the IL4 gene is in agreement with your data. Pawlik et al presented a significantly increased disease activity, the number of swollen and tender joints, and ESR in RA patients with CT and TT genotypes than those with CC genotype. Do you have information about the genotype of the studied cases?
21. Please, discuss the advantages and disadvantages of multiplex assay! Why researchers and clinicians should analyze 27-cytokines, chemokines, and growth factors as a diagnostic tool for effectively separating rheumatoid arthritis patients from healthy controls?
Conclusion:
22. Please, revise the first and second statements of the conclusion (lines 453-455).
References:
23. Please, check the reference list and correct the format according to the journal instructions
Minor issues:
Abstract
24. Page 1, line 33: Please, check what molecule was analysed PDGF-BB or PDGFR?
Introduction:
25. Page 2, line 79: Please, provide a reference for the following statement “PDGF-BB has an important role in RA-FLS proliferation and migration.”
Results:
26. Please, put the tables/figures next to the connected text.
27. All used abbreviations in Tables and Figures should be explained in the legends
28. If you detect autoantibodies to cyclic citrullinated peptides, please correct in text, and figures CCP to anti-CCP.
Author Response
Reviewer 1:
The manuscript titled: “Multivariate diagnostic biomarkers of rheumatoid arthritis.” submitted to the journal Biomolecules with manuscript ID: biomolecules-2535160 by Wail M. Hassan et al. analyzed 27 serological factors by multiplex assay - Bio-Plex assay in a group of 78 cases with rheumatoid arthritis (RA). The authors analyzed data by different statistical tests and demonstrated that using binary logistic regression led to higher fidelity of rheumatoid arthritis patients’ identification than using discriminant analysis.
Although the general aim and scope of the journal are focused on structures and functions of bioactive and biogenic substances, molecular mechanisms with biological and medical implications as well as biomaterials and their applications, with the respect of the Bioinformatics and Systems Biology Section of the journal I accept that the manuscript could be of interest to the journal’s readers.
However, the manuscript has several weaknesses that should be taken into consideration.
Major issues:
Title:
- The title “Multivariate diagnostic biomarkers of rheumatoid arthritis” should be revised.
In my opinion, the main topic of the manuscript is the research for the most trustful and powerful analysis able to discriminate individuals with RA from controls.
Response: We changed the title to “The fidelity of rheumatoid arthritis multivariate diagnostic biomarkers using discriminant analysis and binary logistic regression”.
Abstract:
- Page 1, lines 31-33: Please revise the sentence: “The obtained data showed that a 27-cytokine, multiparametric biomarker profile can successfully distinguish rheumatoid arthritis patients from healthy controls.”
Response: Edited, new sentence: We show that multiple cytokine biomarker profiles successfully distinguished rheumatoid arthritis patients from healthy controls. - Out of 27 tested serological biomarkers, just 4 or 5 of them (IL-4; IL-17; MIP-1b; RANTES; PDGF-BB) showed some significance. This should be clearly stated.
Response: In response to the reviewer comment, we introduced the following statement in the abstract: “IL-17, IL-4, and RANTES were among the most predictive variables and made up both BLR and DA predictive models for pooled participants (men and women). For the women-only models, the significant cytokines incorporated in the model were IL-4, IL-17, MIP-1b, and RANTES for the BLR model, and IL-4, IL-1Ra, GM-CSF, IL-17, and eotaxin for the DA model. The BLR and DA men-only models contained one cytokine each, eotaxin for BLR and PDGF-bb for DA.”.
Materials and Methods:
2.1. Patients and controls
- The small sample size. The study included 78 RA cases 57 of which with available clinical data.
- 5. A detailed table with clinical and demographic data of cases and controls should be added. The current therapies also should be taken into account.
Response:
For both comments a detailed table with clinical and demographic data of cases and controls (Table 1) and all tables were renumbered accordingly in the results and discussion sections.
- It is not clear how many controls were included. What inclusion and exclusion criteria were used for controls?
Response
Done, the number of controls together with the inclusion and exclusion criteria were mentioned, and you can find highlighted in yellow within the text.
- It is not clear for which analyses the RA patients were grouped into high, moderate, and low, and remission groups. The subdivision into 4 subgroups of RA patients probably will influence the power of the analysis.
Response:
Done you can find highlighted in yellow within the text
- A new subheading “Bio-Plex assay” or similar should be added. The procedure does not fit well with 2.2. “Sample acquisition and preparation”. The technical characteristics such as the sensitivity, precision, accuracy, detection range etc. of the assay should be added.
- I may suggest validation of the obtained results, at least for the most significant cytokines, by the singleplex assays, as conventional ELISA tests that may strengthen your results. Multiplex approaches usually are accepted as screening tests, for the future focusing on the most significant parameters.???
Response:
Thanks for your suggested idea, we can consider in our future studies
2.3. Statistical analysis
- This part should be shortened and revised. The used reference in that part did not correspond to the reference list. Please, check all of them carefully. Also, correct the format of references according to the journal instruction.
Response: We corrected the references and properly formatted them. In regard to shortening the description of the statistical methods used, we respectfully disagree with the reviewer. This level of detail is essential for the unfamiliar reader to understand the mathematical differences between, for example, BLR and DA and how each model is evaluated. Furthermore, some of the statistical methods, such as hierarchical clustering, BLR, and DA, can be done different ways, and for the reader to be able to reproduce our data, they must know how we performed such analyses and what choices we made. The similarity coefficient used in cluster analysis is just one example; there are many of those and using a different one can greatly impact the results. Therefore, the detailed description of our stats is inseparable from the remainder of the manuscript. We acknowledge that the lack of such detail is common in the literature, but we advocate for that to change, particularly so in recent years when multivariate methods have become more common and many use them or read their output without proper understanding.
Results and discussion:
- The section “Results and discussion” should be separated into two “Results” and “Discussion”
Response:
- Results and discussion section was separated, you can find highlighted in yellow.
- The observed levels in pg/ml (row data, mean, or median) of studied cytokines, chemokines, and growth factors should be added.
- Figure 2, should be revised. It is not entirely informative. What is presented on the x-axis? Also, please check the legend – the last sentence should be deleted.
Response: We thank the reviewer for this comment. We deleted the unnecessary sentence from the legend. The figure provides a visual representation of how the participants were partially separated using DA or BLR and the degree of overlap in each case. This is relevant to the following figure that shows the rates of correct classification based on these two methods. To help the reader understand the figure we had symbols for male and female and colors (green and red) to distinguish between RA and control. In response to the reviewer comment, we added the following to the legend to improve clarity: “In the discriminant analysis plot (top), the x axis represents the first canonical discriminant function and participants are graphed using their discriminant scores. In the binary logistic regression plot (bottom), the x axis represents the probability of having rheumatoid arthritis.”.
- Table 4 and Figure 4 could be combined into one.
- The observed correlations with clinical parameters even significant are weak and this should be clearly stated.
Response:
- Done and you can find highlighted in yellow within the text
- Please check carefully the data presented in Figure 5 and Figure S1. For me, it is hard to believe that the differences between them are just 2 men, RA patients.
- It was revised and it is correct. Thanks for your care
- Discussion should be a separate part of the paper and should be focused on the obtained results.
Response:
- Done and you can find highlighted in yellow within the text
- How statements as on page 7, lines 310-313 “In blood arteries and cardiomyocytes, IL-17 promotes inflammation, blood coagulation, and thrombosis [25]. It is interesting to mention that Inhibiting IL-17 expression or reducing its interaction with the receptor may thus be beneficial in treating cardiovascular disease in RA patients [26].” is connected with the presented data. If cases with cardiovascular disease are included in the study, please clarify this.
- Please, clarify the statement “…..ESR is a reflection of fibrinogen level in the blood………” (page 10, line 410)
Response:
- Done and the clarified statement was supported by a reference
- For me it is not clear, how the associative study by Pawlik et al [40] concerning promoter polymorphism in the IL4 gene is in agreement with your data.Pawlik et al presented a significantly increased disease activity, the number of swollen and tender joints, and ESR in RA patients with CT and TT genotypes than those with CC genotype. Do you have information about the genotype of the studied cases?
- Please, discuss the advantages and disadvantages of multiplex assay! Why researchers and clinicians should analyze 27-cytokines, chemokines, and growth factors as a diagnostic tool for effectively separating rheumatoid arthritis patients from healthy controls?
Response:
Advantages of multiplex assay together with the concept of the measured variables selection were both mentioned and you can find highlighted in yellow.
Conclusion:
- Please, revise the first and second statements of the conclusion (lines 453-455).
Response:
Both were revised and rewritten
References:
- Please, check the reference list and correct the format according to the journal instructions
Response:
All references were carefully revised
Minor issues:
Abstract
- Page 1, line 33: Please, check what molecule was analysed PDGF-BB or PDGFR?
Response:
- No the correct measured variable are PDGF-BB. You can see in the figure
Introduction:
- Page 2, line 79: Please, provide a reference for the following statement “PDGF-BB has an important role in RA-FLS proliferation and migration.”
Response:
-Done
Results:
- Please, put the tables/figures next to the connected text
- Response:
Done and you can see within the text
- All used abbreviations in Tables and Figures should be explained in the legends
- Already done
- If you detect autoantibodies to cyclic citrullinated peptides, please correct in text, and figures CCP to anti-CCP.
Response:
Done , Please see the demographic and Seropositivity data in Table 1 ( CCP in the table is correct)
Reviewer 2 Report
Comments to the authors
I read this manuscript with interest. These are my comments.
1. I think PDGFR should be spelled out in the Abstract
2. Rheumatoid arthritis and RA should be unified in the manuscript. Sometimes RA, but sometimes rheumatoid arthritis.
3. Are you citing Figure 1 in the text?
4. Line 239, Results and Discussion should be separately written.
5. Please add abbreviations below Table 3
6. Figure 3 Binary Logistic regression, there are a misspelling.
Author Response
Reviewer 2:
I read this manuscript with interest. These are my comments.
1.I think PDGFR should be spelled out in the Abstract
Response:
Done
- Rheumatoid arthritis and RA should be unified in the manuscript. Sometimes RA, but sometimes rheumatoid arthritis.
Response:
- Thanks for your comment. It was unified along the manuscript
- Are you citing Figure 1 in the text?
Response:
- Thank you, it was missing but now we cited.
- Line 239, Results and Discussion should be separately written.
Response:
Done and you can find highlighted in yellow within the text
- Please add abbreviations below Table 3
Response
- All scientific expression are written in full name
- Figure 3 Binary Logistic regression, there are a misspelling.
Response: We found one typo (alogorithm), which was corrected. Thank you!
Round 2
Reviewer 1 Report
The revised version of the manuscript titled: “The fidelity of rheumatoid arthritis multivariate diagnostic biomarkers using discriminant analysis and binary logistic regression” former title “Multivariate diagnostic biomarkers of rheumatoid arthritis.” submitted to the journal Biomolecules with manuscript ID: biomolecules-2535160 by Wail M. Hassan et al. should be revised before publication although authors replied to the most of my previous concerns.
Major issues:
Materials and Methods:
2.1. Patients and controls
1. The sample size is still unclear. Obviously, the logistic regression and discriminant models were constructed from the data of 78 cases and 78 controls tested by Bio-Plex assay. However, the new Table 1 includes 66 cases, on page 13, lines 366-367, the number of cases is 57, the same is noted in legend of the Figure 5, 55 of which were women included in Figure S1. Please, clarify this!
2. Thank you for the new Table 1. Please, include in it the number of RA patients with high, moderate, and low disease severity and in remission. The mean (± SD) and median (interquartile range 25–75) values of DAS28 (ESR), ESR, and anti-CCP also must be included in Table 1.
2.3. Statistical analysis
3. Thank you for your answer! I accept your point of view. Moreover, the main strength of the paper in my opinion is the proof of the binary logistic regression as a test for analysis of multiple biomarkers with better fidelity in identifying rheumatoid arthritis patients than discriminant analysis.
Results:
4. The observed levels in pg/ml (row data, mean, or median) of studied cytokines, chemokines, and growth factors should be added If not all, at least the most significant biomarkers - IL-17, IL-4, RANTES, MIP-1b, IL-1Ra, GM-CSF, eotaxin, and PDGF-BB must be presented in detail.
5. In accordance with your preference for the level of detail in statistical analysis, and with a reverence to the unfamiliar reader to understand them, it is essential to add an explanation that the observed correlations with clinical data are weak, although significant. The Spearman's correlation coefficient r within the range of 0.1-0.39 is regarded as weak, even though it may has a high statistical significance level, it is a weak one.
Discussion
6. The study by Pawlik et al [50] does not correspond to the current study. The associative study by Pawlik et al [50] concerning promoter polymorphism in the IL4 gene presented a significantly increased disease activity, the number of swollen and tender joints, and ESR in RA patients with CT and TT genotypes than those with CC genotype. According to your answer, I accept that you don’t have information about the genotypes of your patients. Here are some parts from the study by Pawlik et al [50].
Pawlik et al. Abstract: “Nevertheless, the active form of RA was more frequently diagnosed in patients with T allele (genotypes CT and TT) as compared with homozygous CC patients. Moreover, in carriers of the T allele, parameters of disease activity (DAS 28 score, ESR, number of swollen and tender joints) were significantly increased. We suggest that the IL-4 )590 promoter polymorphism may be a genetic risk factor for RA severity.”
Pawlik et al. Results, page 50: “…The parameters of disease activity were significantly increased in carriers of T allele as compared with homozygous CC subjects. As shown in Table 3, the number of swollen and tender joints, ESR as well as DAS 28 score were significantly increased in patients with CT and TT genotypes, whereas there were no statistically significant differences with regard to CRP values and morning stiffness duration.”
Conclusion:
7. Conclusion must be revised. As I have mentioned already, the usage of a panel of 27 biomarkers for profiling RA is unnecessary. The presented data showed significance for some of them, according to the model. In addition, the usage of this multiplex assay for 27 biomarkers as a diagnostic tool very likely will lead to an unnecessary increase in the cost. Bearing in mind the price for a 27-plex assay, 1x96 plate, useful for analysis of 38 subjects (76 duplicated samples) it could be assumed that more cost-effective will be a custom 8-plex assay including a panel of the most important biomarkers for the all-participants and sex-specific models.
Minor issues:
8. Please, correct “CCP” to “anti-CCP” all over the manuscript, including text, figures, and tables.
9. Page 16, first paragraph, lines 392-401: Probably this text concern Table 4 which is not referred to in the text. Please, check!
10. The cut-off value for the positivity of RF or anti-CCP should be added.
11. Please, check the reference on page 13, line 359.
12. Please, be more careful with highlighting of the revised version of the manuscript.
Author Response
Point by point response to reviewer’s 1 comments
Comments and Suggestions for Authors:
The revised version of the manuscript titled: “The fidelity of rheumatoid arthritis multivariate diagnostic biomarkers using discriminant analysis and binary logistic regression” former title “Multivariate diagnostic biomarkers of rheumatoid arthritis.” submitted to the journal Biomolecules with manuscript ID: biomolecules-2535160 by Wail M. Hassan et al. should be revised before publication although authors replied to the most of my previous concerns.
Major issues:
Materials and Methods:
2.1. Patients and controls
- The sample sizeis still unclear. Obviously, the logistic regression and discriminant models were constructed from the data of 78 cases and 78 controls tested by Bio-Plex assay. However, the new Table 1 includes 66 cases, on page 13, lines 366-367, the number of cases is 57, the same is noted in legend of the Figure 5, 55 of which were women included in Figure S1. Please, clarify this!
Response:
- The clinical data was only available for 57 out of the 78 RA patients, the number 66 in the table was wrong and now it is corrected. This was clearly written below table 1. All percentages of the available clinical data presented in table 1 were corrected accordingly.
- Thank you for the new Table 1. Please, include in it the number of RA patients with high, moderate, and low disease severity and in remission. The mean (± SD) and median (interquartile range 25–75) values of DAS28 (ESR), ESR, and anti-CCP also must be included in Table 1.
Response:
- Sorry as we don’t have access to the clinical data now but we can consider your valuable comments in our future studies
2.3. Statistical analysis
- Thank you for your answer! I accept your point of view. Moreover, the main strength of the paper in my opinion is the proof of the binary logistic regression as a test for analysis of multiple biomarkers with better fidelity in identifying rheumatoid arthritis patients than discriminant analysis.
Response:
- Thanks and this was highlighted in the results and conclusion
Results:
- The observed levels in pg/ml (row data, mean, or median) of studied cytokines, chemokines, and growth factors should be added If not all, at least the most significant biomarkers - IL-17, IL-4, RANTES, MIP-1b, IL-1Ra, GM-CSF, eotaxin, and PDGF-BB must be presented in detail.
- The Whole complete excel sheet of the data was uploaded as supplementary materials
- In accordance with your preference for the level of detail in statistical analysis, and with a reverence to the unfamiliar reader to understand them, it is essential to add an explanation that the observed correlations with clinical data are weak, although significant. The Spearman's correlation coefficient r within the range of 0.1-0.39 is regarded as weak, even though it may has a high statistical significance level, it is a weak one.
- Response:
In relation to your comment the results section was rewritten to be clearer for the readers. You can find highlighted in cyane within the text.
Discussion
- The study by Pawlik et al [50] does not correspond to the current study. The associative study by Pawlik et al [50] concerning promoter polymorphism in the IL4 gene presented a significantly increased disease activity, the number of swollen and tender joints, and ESR in RA patients with CT and TT genotypes than those with CC genotype. According to your answer, I accept that you don’t have information about the genotypes of your patients. Here are some parts from the study by Pawlik et al [50].
Pawlik et al. Abstract: “Nevertheless, the active form of RA was more frequently diagnosed in patients with T allele (genotypes CT and TT) as compared with homozygous CC patients. Moreover, in carriers of the T allele, parameters of disease activity (DAS 28 score, ESR, number of swollen and tender joints) were significantly increased. We suggest that the IL-4 )590 promoter polymorphism may be a genetic risk factor for RA severity.”
Pawlik et al. Results, page 50: “…The parameters of disease activity were significantly increased in carriers of T allele as compared with homozygous CC subjects. As shown in Table 3, the number of swollen and tender joints, ESR as well as DAS 28 score were significantly increased in patients with CT and TT genotypes, whereas there were no statistically significant differences with regard to CRP values and morning stiffness duration.”
Response:
- Done, the whole paragraph was rewritten and supported through considering more relevant work. Kindly you can find highlighted in cyane (Lines 479-497).
Conclusion:
- Conclusion must be revised. As I have mentioned already, the usage of a panel of 27 biomarkers for profiling RA is unnecessary. The presented data showed significance for some of them, according to the model. In addition, the usage of this multiplex assay for 27 biomarkers as a diagnostic tool very likely will lead to an unnecessary increase in the cost. Bearing in mind the price for a 27-plex assay, 1x96 plate, useful for analysis of 38 subjects (76 duplicated samples) it could be assumed that more cost-effective will be a custom 8-plex assay including a panel of the most important biomarkers for the all-participants and sex-specific models.
Response:
- Thanks, the conclusion was rewritten according to your raised comment
Minor issues:
- Please, correct “CCP” to “anti-CCP” all over the manuscript, including text, figures, and tables.
-Response:
Done and corrected along the manuscript
- Page 16, first paragraph, lines 392-401: Probably this text concern Table 4 which is not referred to in the text. Please, check!
Response:
- Done and Table 4 was referred to 3 times within the text
- The cut-off value for the positivity of RF or anti-CCP should be added.
-Response: Done and you can find highlighted within Table 1
- Please, check the reference on page 13, line 359.
Response:
- All references were carefully checked and you can find highlighted in red within the text
- Please, be more careful with highlighting of the revised version of the manuscript.
- We hope that we carefully highlighted the revised version and apologize for the inconvenience